# Preclinical Study of Pharmacokinetic/Pharmacodynamic Analysis of Tebipenem Using Monte Carlo Simulation for Extended-Spectrum β-Lactamase-Producing Bacterial Urinary Tract Infections in Japanese Patients According to Renal Function

**DOI:** 10.3390/antibiotics14070648

**Published:** 2025-06-26

**Authors:** Fumiya Ebihara, Takumi Maruyama, Hidefumi Kasai, Mitsuru Shiokawa, Nobuaki Matsunaga, Yukihiro Hamada

**Affiliations:** 1Department of Pharmacy, Tokyo Women’s Medical University Hospital, Tokyo 162-8666, Japan; ebihara.fumiya@twmu.ac.jp (F.E.); shiokawa.mitsuru@twmu.ac.jp (M.S.); 2Department of Pharmacy, Kochi Medical School Hospital, Kochi 783-8505, Japan; jm-maruyama.takumi@kochi-u.ac.jp; 3Keio Frontier Research and Education Collaboration Square (K-FRECS) at Tonomachi, Keio University, Kawasaki 210-0821, Japan; 4AMR Clinical Reference Center, National Center for Global Health and Medicine Japan Institute for Health Security, Tokyo 162-8655, Japan; matsunaga.no@jihs.go.jp

**Keywords:** tebipenem, extended-spectrum β-lactamase, urinary tract infection, pharmacokinetics/pharmacodynamics, renal function, Monte Carlo simulation

## Abstract

Background/Objectives: The increasing prevalence of urinary tract infections (UTIs) caused by extended-spectrum β-lactamase (ESBL)-producing organisms poses a significant clinical challenge worldwide due to limited oral treatment options. Tebipenem (TBPM), an oral carbapenem antibiotic, is currently approved only for pediatric use in Japan, with no adult indication established. International studies have shown promising results for ESBL-producing infections, but optimal dosing regimens for Japanese adults with varying renal function have not been established. This study aimed to determine the optimal TBPM dosing regimens for ESBL-producing Enterobacterales UTIs in Japanese patients stratified by renal function, providing evidence for potential adult approval applications in Japan. Methods: Monte Carlo simulations (MCSs) were performed using pharmacokinetic parameters derived from clinical trials in Japanese subjects. Various dosing regimens were evaluated across different creatinine clearance (CCR) ranges and minimum inhibitory concentrations (MICs). The pharmacokinetic/pharmacodynamic target was set at *f*AUC_0–24_/MIC·1/tau ≥ 34.58, with a ≥90% probability of target attainment (PTA) considered optimal. Results: For patients with severe renal impairment (CCR < 30 mL/min), 150 mg q12 h achieved a >90% PTA against ESBL-producing organisms with an MIC of 0.03 mg/L. For moderate-to-severe renal impairment (30 ≤ CCR < 50 mL/min) and moderate renal impairment (50 ≤ CCR < 80 mL/min), 300 mg q8 h maintained a >90% PTA. For normal renal function (CCR ≥ 80 mL/min), 600 mg q8 h was required to achieve the target PTA. Conclusions: This first Japanese PK/PD analysis of TBPM in ESBL-producing UTIs provides evidence-based dosing recommendations across various renal function levels. TBPM, with appropriate renal-adjusted dosing, may offer an effective oral treatment option for patients who have traditionally required hospitalization for parenteral therapy.

## 1. Introduction

In recent years, urinary tract infections (UTIs) caused by extended-spectrum β-lactamase (ESBL)-producing organisms have been increasing worldwide, posing a significant public health challenge [1]. In Japan, the clinical and economic burden of UTIs is considerable, with an estimated 106,508 annual hospitalizations for UTIs and a median medical cost of USD 4250 per hospitalization [2]. According to the Fourth National Japanese Antimicrobial Susceptibility Pattern Surveillance Program conducted from July 2020 to December 2021, *E. coli* was identified in 54.3% of complicated UTIs (cUTIs), with ESBL-producing *E. coli* strains accounting for 24.8% of all *E. coli* strains, showing a sustained increase over time (5.1% in 2008, 15.2% in 2011, and 24.3% in 2015) [3]. Similarly, ESBL-producing *K. pneumoniae* strains have shown comparable trends in resistance patterns. This trend represents a significant healthcare challenge that urgently requires effective oral treatment options. Meta-analyses have shown that infections caused by ESBL-producing organisms are associated with a 2.35-fold higher mortality compared with those caused by non-ESBL-producing organisms [4], as well as with prolonged hospitalization, increased healthcare costs, and reduced treatment response rates [5,6,7]. The main cause of poor prognosis is inappropriate initial antibiotic therapy, with mortality rates of 3.7% when treated appropriately with carbapenems versus 44–64% with inappropriate antibiotic selection [8]. Therefore, appropriate antibiotic selection and dosing regimens are crucial for ESBL-producing bacterial infections. However, treatment options are limited due to resistance to multiple antibiotics, making outpatient treatment with oral antibiotics particularly challenging.

Tebipenem (TBPM) is a carbapenem antibiotic, and tebipenem pivoxil hydrobromide (TBPM-PI-HBr), a prodrug of its active moiety, is the world’s first orally available carbapenem antibiotic with enhanced oral absorption [9]. TBPM has been reported to exhibit excellent antimicrobial activity against many Gram-negative and Gram-positive bacteria, including ESBL-producing organisms [10,11]. In Japan, TBPM is currently approved only for pediatric use at a dose of 4 mg/kg twice daily after meals (as tebipenem pivoxil), with a maximum dose that can be increased up to 6 mg/kg as needed [12], and has no established adult application. Given the significant clinical and economic burden of UTIs requiring hospitalization, effective oral carbapenem therapy, such as TBPM, could provide substantial healthcare benefits. Assuming that using an effective oral carbapenem could reduce hospitalizations for complicated UTIs by 10%, it is projected that approximately 510,000 hospital bed-days could be saved annually, leading to a reduction in total medical costs of USD 45 million [13].

Recently, international clinical trials of TBPM-PI-HBr for adult complicated urinary tract infections have been conducted, with a randomized comparative trial (ADAPT-PO trial) demonstrating non-inferiority compared to ertapenem injection [14]. These results suggest that oral therapy may become a new treatment option for patients with complicated urinary tract infections who have traditionally required hospitalization for injectable therapy. However, the optimal dosage for adult patients with varying degrees of renal function has not yet been clearly established, and, internationally, appropriate dosing regimens based on renal function have not been standardized. Although dose adjustment based on renal function has been suggested [15], there are limited studies on dose optimization based on pharmacokinetic parameters and pharmacokinetic/pharmacodynamic (PK/PD) simulations specifically for Japanese adult patients.

Recently, Ganesan et al. developed a population pharmacokinetic model using international clinical trial data, which incorporated various patient factors, including renal function, body size, and infection status [16]. However, potential ethnic differences in pharmacokinetic characteristics between Japanese and non-Japanese populations may necessitate population-specific dosing approaches.

With the increase in urinary tract infections caused by ESBL-producing bacteria and the urgent need for oral treatment options for adults in Japan, this study was designed to support regulatory approval applications by providing evidence-based dosing recommendations. The objective of this study was to evaluate the appropriate dosage of TBPM and clinical PK/PD breakpoints for UTIs caused by ESBL-producing Enterobacterales in Japanese adult patients, considering renal function. Additionally, we compared our Japanese model with the international population pharmacokinetic model to validate the need for population-specific dosing strategies. This research aims to provide the preclinical evidence necessary for adult TBPM approval in Japan.

## 2. Results

To evaluate the probability of target attainment (PTA) for different TBPM dosing regimens against ESBL-producing *E. coli* and *K. pneumoniae* with varying minimum inhibitory concentrations (MICs), Monte Carlo simulations (MCSs) were conducted across various creatinine clearance (CCR) ranges. Figure 1 shows the PTA results for different dosing regimens, CCR ranges, and MICs. For patients with severe renal impairment (CCR < 30 mL/min), all dosing regimens achieved a >90% PTA against organisms with an MIC of 0.03 mg/L. Even the 150 mg q12 h regimen achieved a PTA of 97.3%. For patients with moderate-to-severe renal impairment (30 ≤ CCR < 50 mL/min), the 150 mg q12 h and 250 mg q12 h regimens failed to achieve the target, with PTAs of 40.6% and 74.9%, respectively. However, both the 300 mg q8 h and 600 mg q8 h regimens achieved the target, with PTAs of 99.4% and 100%, respectively.

For patients with moderate renal impairment (50 ≤ CCR < 80 mL/min), the 150 mg q12 h and 250 mg q12 h regimens achieved PTAs of only 7.4% and 32.3%, respectively, against organisms with an MIC of 0.03 mg/L. The 300 mg q8 h regimen maintained robust efficacy, with a PTA of 93.3%, while the 600 mg q8 h regimen provided nearly complete coverage, with a PTA of 99.7%. For organisms with an MIC of 0.06 mg/L, only the 300 mg q8 h (56.4% PTA) and 600 mg q8 h (93.3% PTA) regimens provided significant coverage.

Finally, for patients with normal renal function (CCR ≥ 80 mL/min), only the 600 mg q8 h regimen achieved the target PTA, with a rate of 98.4%.

Table 1 summarizes the recommended TBPM dosing regimens by renal function level for the optimal treatment strategy against ESBL-producing *E. coli* and *K. pneumoniae* (MIC 0.03 mg/L), as derived from the MCS results.

To assess population-specific dosing requirements, we compared our Japanese model with the Ganesan population pharmacokinetic model using standardized simulation conditions (fasting state, infected patients, and identical renal function categories and dosing regimens). As shown in Figure 2, our Japanese model consistently predicted higher free AUC_0–24_ values compared to the Ganesan model across all renal function groups and dosing regimens. When examining the recommended dosing regimens from Table 1, the Japanese model shows progressively higher exposures with declining renal function: 1.6-fold higher in patients with normal renal function (CCR ≥ 80 mL/min) receiving 600 mg q8 h; 1.8-fold higher in moderate renal impairment (50 ≤ CCR < 80 mL/min) with 300 mg q8 h; 2.2-fold higher in moderate-to-severe renal impairment (30 ≤ CCR < 50 mL/min) with 300 mg q8 h; and 1.5-fold higher in severe renal impairment (CCR < 30 mL/min) with 150 mg q12 h. These differences in drug exposure translate to higher PTA values in the Japanese model, as summarized in Table 2. Notably, our recommended dosing regimens differed from those that would be suggested by the Ganesan model. For patients with moderate renal impairment (50 ≤ CCR < 80 mL/min), the Japanese model achieved a >90% PTA with 300 mg q8 h (93.3%), while the Ganesan model would require 600 mg q8 h to reach a similar efficacy. For patients with normal renal function (CCR ≥ 80 mL/min), the Japanese model achieved a 98.4% PTA with 600 mg q8 h, whereas the Ganesan model achieved a 87.9% PTA with the same regimen. These findings suggest distinct dosing requirements between Japanese and international populations.

## 3. Discussion

To our knowledge, as TBPM remains an unapproved drug for adult use in Japan, this study represents the first Japanese evaluation of appropriate TBPM dosing regimens based on renal function for urinary tract infections caused by ESBL-producing Enterobacterales. Internationally, the ADAPT-PO trial conducted by Eckburg et al. demonstrated that TBPM is non-inferior to ertapenem injection for the treatment of complicated urinary tract infections and acute pyelonephritis [14]. This trial compared oral tebipenem with intravenous ertapenem in 868 hospitalized patients and confirmed no significant difference in overall efficacy rates at the time of cure assessment (58.8% vs. 61.6%; weighted difference: −3.3%). These results suggest that oral therapy may become a new treatment option for patients who have traditionally required hospitalization for injectable therapy. Interestingly, in the ADAPT-PO trial, oral TBPM administration showed a high overall efficacy rate of 93.6% at the end of treatment in patients with bacteremia [14]. This finding suggests that oral TBPM at appropriate dosages may be effective even in severe infections traditionally considered to require intravenous antibiotics. Regarding the PK/PD of TBPM, McEntee et al. identified *f*AUC_0–24_/MIC·1/tau as the most appropriate indicator and established a target value of ≥34.58 for bactericidal effects and resistance suppression [10]. Our study uniquely evaluated optimal dosing regimens at different levels of renal function in the Japanese patient population based on this target value.

For our MCSs, because there have been no population pharmacokinetic analyses of TBPM in Japanese adults, we utilized the pharmacokinetic parameters reported by Nakashima et al. [15] and applied inter-individual variability values from the faropenem (FRPM) studies of Hirooka et al. [17]. To the best of our knowledge, there are no other oral carbapenems, nor are there any data on the population pharmacokinetics of TBPM in Japanese adults. Therefore, we used the pharmacokinetic parameter of FRPM as an alternative, as it was considered to have a similar elimination route among oral penems. The inter-individual variability values used from FRPM (ωCL/F = 53%; ωVd/F = 37%; ωka = 67%) are comparable to those recently reported in Ganesan et al.’s population pharmacokinetic model of TBPM (ωCL/F (phase 3) = 57.2%; ωVc/F = 44.4%; ωka = 71.9%) [16], providing retrospective support for our approximation approach. Nevertheless, if the actual variability in TBPM pharmacokinetics differs substantially from our assumptions, our PTA estimates could be affected. For instance, if the true ωVd/F for TBPM were higher than our assumed 37%, patients would show greater variability in drug distribution, potentially leading to a larger proportion of patients with subtherapeutic exposure and, consequently, lower PTA values. However, variations in ωka would primarily affect the rate of absorption rather than the extent of absorption and would, therefore, not be expected to significantly impact systemic AUC, which is the relevant measure of tebipenem exposure for efficacy. The adoption of the FRPM variability parameters represents a pragmatic approach given the data limitations, but it is an acknowledged limitation of our study.

To validate our Japanese model approach, we compared it with the population pharmacokinetic model recently developed by Ganesan et al. using data from international subjects [16]. Their model utilized a two-compartment structure with linear, first-order elimination and two transit compartments to describe drug absorption, identifying several clinical factors affecting tebipenem pharmacokinetics, including renal function, body size, and infection status. We conducted comparative simulations using the Ganesan model with standardized demographics (body surface area (BSA): 1.86 m^2^; height: 169 cm) representing international population averages, while our Japanese model reflected the actual demographic characteristics of Japanese subjects (mean BSA: 1.67 m^2^; mean body weight: 61.9 kg; mean height: 164.3 cm). Both models used identical simulation conditions, including fasting state, infected patients, and Phase 3 parameters, and the same renal function categories and dosing regimens. We then compared the resulting AUC distributions and the probability of target attainment (Figure 2 and Table 2).

As shown in Figure 2, our Japanese model predicted consistently higher AUC_0–24_ values compared with the Ganesan model across all renal function groups. When examining the recommended dosing regimens from Table 1, the Japanese model showed progressively higher exposures with declining renal function: 1.6-fold higher in patients with normal renal function (CCR ≥ 80 mL/min) receiving 600 mg q8 h; 1.8-fold higher in moderate renal impairment (50 ≤ CCR < 80 mL/min) with 300 mg q8 h; 2.2-fold higher in moderate-to-severe renal impairment (30 ≤ CCR < 50 mL/min) with 300 mg q8 h; and 1.5-fold higher in severe renal impairment (CCR < 30 mL/min) with 150 mg q12 h. These differences in exposure translated to higher PTA values in the Japanese model, as shown in Table 2. For example, at an MIC of 0.03 mg/L with the 300 mg q8 h regimen, the Japanese model predicted a 93.3% PTA in the moderate renal impairment group (50 ≤ CCR < 80 mL/min) compared to the 62.2% of the Ganesan model. These findings suggest that the demographic differences between Japanese and international populations, particularly the smaller body size of Japanese patients (61.9 kg vs. 76 kg body weight; 1.67 m^2^ vs. 1.86 m^2^ BSA), combined with different clearance modeling approaches, result in higher tebipenem exposures per dose in Japanese patients, supporting population-specific dosing approaches. The comparison between our Japanese and international models provides important clinical practice implications for different healthcare settings. Given the differences in patient demographics and clearance characteristics, dosing strategies developed for international populations require adjustment for Japanese clinical practice. These differences indicate that optimal dosing regimens identified through our Japanese model may not translate directly to international clinical settings, where higher doses could be necessary to achieve therapeutic targets. These demographic-driven pharmacokinetic differences have practical implications for dose selection in clinical practice. For example, in elderly Japanese patients with moderate renal impairment, our study demonstrates that 300 mg q8 h achieves adequate therapeutic exposure, whereas international guidelines based on Western populations might suggest higher doses that could lead to unnecessary drug exposure. Similarly, in Japanese patients with severe renal impairment (CCR < 30 mL/min), our model supports the efficacy of 150 mg q12 h, potentially avoiding overdosing that might occur with the direct application of international recommendations. This emphasizes the importance of population-specific clinical validation before implementing dosing regimens across different ethnic groups.

Regarding dialysis-dependent patients, previous pharmacokinetic studies have provided important insights into TBPM behavior during hemodialysis. Patel et al. conducted a comprehensive study in patients with end-stage renal disease receiving hemodialysis and found that the TBPM pharmacokinetics were significantly altered compared to those in patients with normal renal function [18]. In their study, patients with end-stage renal disease had an approximately 7-fold increase in AUC and elimination half-life (t1/2) compared to those with normal renal function. During a 4 h hemodialysis session, mean TBPM plasma exposure decreased by approximately 40%, with an extraction ratio of 41.1% ± 3.7% and estimated hemodialysis clearance of 8.6 ± 0.78 L/h, demonstrating that TBPM is effectively removed by hemodialysis [19]. These findings suggest that dosing adjustments and timing considerations relative to hemodialysis sessions would be important for optimizing therapy in this patient population. However, specific dosing recommendations for dialysis patients require further clinical validation, and comprehensive guidelines for patients on peritoneal dialysis have not been established.

Recent studies have raised concerns about the potential for FRPM resistance development and possible cross-resistance to carbapenems [1]. Gandra et al. demonstrated that the development of resistance to FRPM through serial passage resulted in cross-resistance to carbapenem antibiotics in certain ESBL-producing isolates [1]. While our study focuses on TBPM rather than FRPM, these findings highlight the importance of appropriate dosing strategies for oral penems and carbapenems.

Our choice of pharmacodynamic targets for ESBL-producing Enterobacterales was supported by a comprehensive analysis of domestic studies on antimicrobial susceptibility patterns. In Japanese surveillance studies conducted between 1998 and 2003, Muratani et al. reported that for ESBL-producing *E. coli* strains from Japanese medical institutions, tebipenem showed MIC50 and MIC90 values of 0.016 and 0.03 mg/L, respectively. For ESBL-producing *K. pneumoniae* (26 strains), tebipenem demonstrated MIC50 and MIC90 values of 0.016 and 0.016 mg/L, respectively. Notably, tebipenem inhibited the growth of all strains at 0.03 mg/mL and was not affected by ESBL production [19]. This is consistent with more recent Japanese data from Ito et al., who collected 229 non-ESBL-producing and 61 ESBL-producing *E. coli* strains from 5 major hospitals in the Gifu Prefecture in 2019. The MIC50 and MIC90 values of TBPM for non-ESBL and ESBL-producing *E. coli* were ≤0.03 μg/mL, with MIC ranges of ≤0.03–0.25 μg/mL and ≤0.03–0.06 μg/mL, respectively, indicating strong antimicrobial activity. Notably, no strains demonstrated reduced susceptibility to TBPM. These results are comparable to those of intravenous carbapenems [13].

International studies have reported somewhat different MIC distributions. Mendes et al. evaluated tebipenem against Enterobacterales causing UTIs in US medical centers (2019–2020) and reported MIC90 values of 0.015 to 0.03 μg/mL against *E. coli* and *K. pneumoniae*. For ESBL-producing isolates (351 *E. coli*, 81 *K. pneumoniae*, and 10 *P. mirabilis* displaying MIC results of ≥2 μg/mL for ceftazidime, aztreonam, and/or ceftriaxone), TBPM showed MIC50 and MIC90 values of 0.015 and 0.06 mg/L, respectively [20]. Similarly, Gerges et al. reported that for ESBL-producing *E. coli* isolated from cancer patients, TBPM showed MIC50 and MIC90 values of 0.015 and 0.03 mg/L, respectively. For ESBL-producing *K. pneumoniae* isolates, TBPM demonstrated MIC50 and MIC90 values of 0.03 and 0.125 mg/L, respectively [21]. Although a large quantity of data is needed to accurately determine the susceptibility of ESBL-producing *E. coli* and *K. pneumoniae* in Japan, the domestic surveillance data consistently support our selection of 0.03 mg/L as a representative MIC90 value for our pharmacokinetic/pharmacodynamic simulations, with the MICs evaluation ranging from 0.008 to 2 mg/L to account for potential variability.

Our study recommends q8 h dosing for patients with a creatinine clearance of ≥30 mL/min, which presents important practical considerations for outpatient care. The requirement for three-times-daily dosing may present adherence challenges, as multiple studies have demonstrated that an increased dosing frequency is associated with decreased medication adherence.

The Japanese COSMOS study (*n* = 1068) demonstrated an overall compliance rate of 74.7% with oral antibiotics. Importantly, compliance was significantly higher, with fewer doses per day and shorter treatment duration. Multivariate analysis identified that the number of doses per day was a statistically significant factor associated with compliance, confirming that prescribing drugs with minimal daily doses represents an effective strategy for improving adherence [22]. International studies consistently support these findings regarding dosing frequency. Llor et al. found that compliance rates decreased significantly with higher dosing frequencies, with mean container openings ranging from 94.3% with once-daily antibiotics to 74.8% with thrice-daily drugs (*p* < 0.001). Only 55.1% of patients in the thrice-daily group took at least 80% of medications, significantly less than those receiving twice-daily (71.4%) or once-daily (86.7%) regimens. The rate of compliance was particularly low when antibiotics were administered thrice daily and in regimens of 7 days or more [23]. The electronically monitored adherence in skin infections (57%) significantly differs from patient-reported adherence (96%), with poor adherence independently associated with poor clinical outcomes [24]. Studies using validated questionnaires have identified that concerns about side effects (37%) and swallowing difficulties (19%) are primary barriers to antibiotic adherence [25]. To address these challenges in Japanese clinical practice, a comprehensive approach should integrate patient education emphasizing the completion of the full antibiotic course with clear explanations of the treatment’s purpose and significance (as recommended by the COSMOS study), leverage Japan’s frequent patient–physician interactions for ongoing adherence monitoring, and implement pharmacist-led interventions that capitalize on Japan’s robust pharmacy network for medication counseling and follow-up support. The clinical implementation of our dosing recommendations must balance optimal pharmacokinetic exposure with practical adherence considerations. While q8 h dosing provides superior efficacy against ESBL-producing organisms, clinicians should consider patient-specific factors, including treatment understanding, duration of therapy, and the documented inverse relationship between dosing frequency and compliance, when implementing these regimens in Japanese clinical practice.

Our study has several limitations. The pharmacokinetic data for TBPM included a limited number of cases in the renal impairment groups (particularly the severe renal impairment group, with only three patients, and the moderate-to-severe renal impairment group, with only two patients), which may affect the reliability of parameter estimation in these groups [15]. Similarly, the application of inter-individual variability values from FRPM to TBPM assumes comparable population variability between these structurally different antimicrobials, which may not be entirely accurate. Additionally, our Japanese model utilized inter-individual variability parameters derived from FRPM studies rather than tebipenem-specific data, while the Ganesan population pharmacokinetic model incorporated variability parameters from actual TBPM clinical trials. This difference in variability modeling may influence the comparison of AUC distributions and PTA calculations between the two models. Clinical patient backgrounds, such as the impact of anatomical and functional urinary tract disorders on treatment efficacy, have not been evaluated, and analysis of the relationship between patient background factors and treatment failure is lacking.

This study primarily focused on pharmacokinetic analysis, with limited evaluation of the relationship with actual clinical and microbiological effects. In particular, there is a lack of long-term follow-up data after the end of treatment and of information on the risk of recurrence, and the direct applicability of international research results to clinical practice in Japan should be verified with the results of clinical trials. Another important consideration is the potential for the development of antimicrobial resistance during TBPM therapy. While TBPM currently demonstrates excellent activity against ESBL-producing organisms [10,11], the selective pressure exerted by their widespread use could potentially lead to the emergence of resistant strains. The lessons from FRPM use in Japan and India are instructive; increased consumption of FRPM has been associated with concerns about rising resistance [1]. While this study focused on plasma pharmacokinetic/pharmacodynamic relationships, recent evidence suggests that urinary antibiotic concentrations may be equally important for predicting microbiological outcomes in cUTI treatment. Melnick et al. demonstrated that differences in urinary clearance between carbapenems significantly impact the timing of recurrent bacteriuria, with ertapenem maintaining inhibitory concentrations (>0.12 μg/mL) for up to 5 days compared to tebipenem’s rapid clearance within 24–36 h [26]. This temporal difference may confound microbiological response assessments in clinical trials. Supporting this concept, Bhavnani et al. found that patient factors, such as urinary tract anatomical disorders and functional abnormalities, were more predictive of treatment failure than plasma antibiotic exposure in the ADAPT-PO trial [27]. Their multivariable analysis showed that anatomical disorders reduced the successful microbiological response by 29–31%, while meaningful relationships between plasma pharmacokinetic indices and efficacy endpoints were not identified [27]. These findings suggest that optimal dosing strategies for urinary tract infections should incorporate both plasma and urinary pharmacokinetic considerations. Future tebipenem dosing optimization studies should include urinary pharmacokinetic modeling to better understand the relationship between local drug exposure and microbiological outcomes, particularly in patients with underlying genitourinary abnormalities. Despite these limitations, the renal function-based dosing guidelines demonstrated in this study may contribute to the optimization of treatment for urinary tract infections caused by ESBL-producing bacteria in Japanese clinical settings. Thus, the findings of this study contribute to the promotion of appropriate antibiotic use.

## 4. Materials and Methods

### 4.1. Pharmacokinetic Parameters

The PK parameters of TBPM were based on results reported by Nakashima M et al. [15]. To our knowledge, there is no population pharmacokinetic model of TBPM for Japanese subjects; therefore, we used mean pharmacokinetic parameter values obtained from clinical trials [15]. The Japanese study population (*n* = 17) had the following demographic characteristics: mean height, 164.3 cm; mean body weight, 61.9 kg; and mean BSA, 1.67 m^2^. Inter-individual variability values were derived from FRPM data reported by Hirooka H et al. Table 3 shows the PK model and parameters used in this study. A one-compartment model with first-order absorption was employed. Table 3 presents the pharmacokinetic parameters obtained from clinical trials of TBPM in Japanese subjects. The inter-individual variability values were derived from FRPM pharmacokinetic data reported by Hirooka et al. [17] and applied to the apparent total body clearance (CL/F), apparent volume of distribution (Vd/F), and absorption rate constant (ka). The unbound fraction of TBPM was set at 0.33 [28].

MCSs were performed to generate blood concentration profiles for 1000 virtual patients for each renal function category, and analyses were conducted using Phoenix™ version 8.3 (Certara, Princeton, NJ, USA). Renal function was evaluated in the following four groups: CCR ≥ 80 mL/min, 50 ≤ CCR < 80 mL/min, 30 ≤ CCR < 50 mL/min, and CCR < 30 mL/min. Dosing regimens of 150 mg (q12 h), 250 mg (q12 h), 300 mg (q8 h), and 600 mg (q8 h) were evaluated.

### 4.2. Pharmacodynamic Data

ESBL-producing *E. coli* and *K. pneumoniae* were selected as target organisms. Based on domestic and international studies [19,20], the MIC90 of TBPM against these strains has been reported to be 0.03 mg/L. In this study, MICs ranging from 0.008 to 2 mg/L were evaluated.

### 4.3. Pharmacokinetic/Pharmacodynamic Parameters and Analysis

TBPM exhibits time-dependent pharmacodynamics, and *f*AUC_0–24_/MIC·1/tau (the ratio of the area under the concentration–time curve of the unbound drug to MIC adjusted for dosing interval) has been shown to be the most appropriate pharmacodynamic indicator [10]. A target value of *f*AUC_0–24_/MIC·1/tau ≥ 34.58 was established as an indicator capable of producing logarithmic bactericidal effects and resistance suppression [10]. The probability of PTA was calculated across MICs for each dosing regimen, with a PTA of ≥ 90% considered as the threshold for optimal dosing.

### 4.4. Comparison with Ganesan Population Pharmacokinetic Model

To validate our Japanese model approach and assess the need for population-specific dosing strategies, we compared our results with the population pharmacokinetic model recently developed by Ganesan et al. [16].

For comparative simulations using the Ganesan model, we applied the following standardized conditions:Population: Infected patients in fasting state using Phase 3 parameters.Demographics: Fixed values of BSA 1.86 m^2^ and height 169 cm (representing international population averages).Protein binding: Unbound fraction of 0.33 (identical to the Japanese model).Renal function groups: Four identical CCR categories (CCR < 30, 30 ≤ CCR < 50, 50 ≤ CCR < 80, and CCR ≥ 80 mL/min).Dosing regimens: Identical dosing schedules (150 mg q12 h, 250 mg q12 h, 300 mg q8 h, and 600 mg q8 h).Simulation conditions: A total of 1000 virtual patients per renal function group using MCS.CCR simulation approach: For the Ganesan model simulations, we used the same average CCR values derived from the Japanese patient data (specifically, 9.1 mL/min (*n* = 3), 40.2 mL/min (*n* = 2), 65.8 mL/min (*n* = 6), and 106.3 mL/min (*n* = 6) for the four renal function categories) [15].Comparison endpoints: Free AUC_0–24_ distributions and PTA values.

In contrast, our Japanese model reflected the demographic characteristics observed in the Phase 1 study by Nakashima et al. [15], where Japanese subjects demonstrated smaller BSA (mean: 1.67 m^2^) and lower body weight (mean: 61.9 kg) compared to the Ganesan model population (median BSA, 1.86 m^2^; median body weight, 76 kg; median height, 169 cm), despite similar height values (mean, 164.3 cm).

An important methodological difference was observed in clearance calculation approaches. Our Japanese model used CCR group-specific mean CL/F values, while the Ganesan model incorporated more complex relationships, including body surface area effects and separate renal and non-renal clearance components with sigmoidal functions.

## 5. Conclusions

This study elucidated the optimal dosage of TBPM for urinary tract infections caused by ESBL-producing Enterobacterales according to renal function through PK/PD analysis. The optimal dosing regimens by renal function were determined to be 600 mg q8 h for normal renal function (CCR ≥ 80 mL/min), 300 mg q8 h for moderate renal impairment (50 ≤ CCR < 80 mL/min), 300 mg q8 h for moderate-to-severe renal impairment (30 ≤ CCR < 50 mL/min), and 150 mg q12 h for severe renal impairment (CCR < 30 mL/min). TBPM, an oral carbapenem antibiotic, may become a new treatment option for complicated urinary tract infection patients who have traditionally required hospitalization for injectable therapy. These preclinical PK/PD study results provide essential evidence-based dosing recommendations that can support future regulatory approval applications for adult TBPM use in Japan, where the drug is currently approved only for pediatric patients. The population-specific dosing guidelines established through this simulation analysis may serve as foundational pharmacokinetic data for potential adult TBPM development in Japanese patients while also contributing to the global understanding of appropriate renal function-based dosing strategies.

## Figures and Tables

**Figure 1 antibiotics-14-00648-f001:**
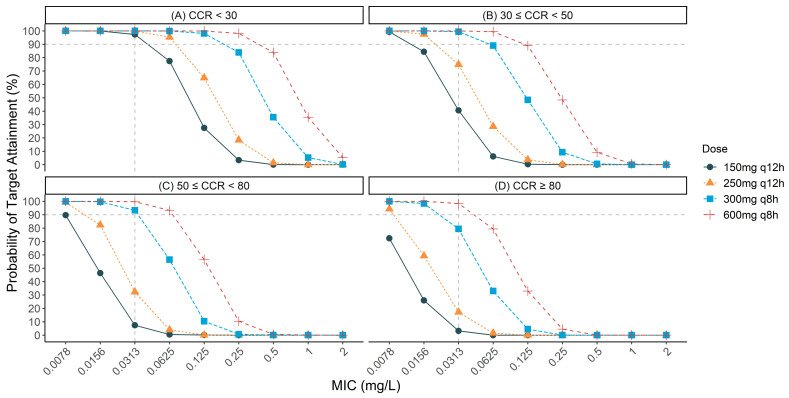
Probability of target attainment (PTA) for various tebipenem dosing regimens against extended-spectrum β-lactamase (ESBL)-producing organisms with different minimum inhibitory concentrations (MICs) across different renal function groups: (**A**) creatinine clearance (CCR) < 30 mL/min; (**B**) 30 ≤ CCR < 50 mL/min; (**C**) 50 ≤ CCR < 80 mL/min; (**D**) CCR ≥ 80 mL/min. The dashed horizontal line represents the 90% PTA threshold considered optimal for clinical efficacy. All the simulations were based on the pharmacokinetic/pharmacodynamic target of *f*AUC_0–24_/MIC·1/tau ≥ 34.58.

**Figure 2 antibiotics-14-00648-f002:**
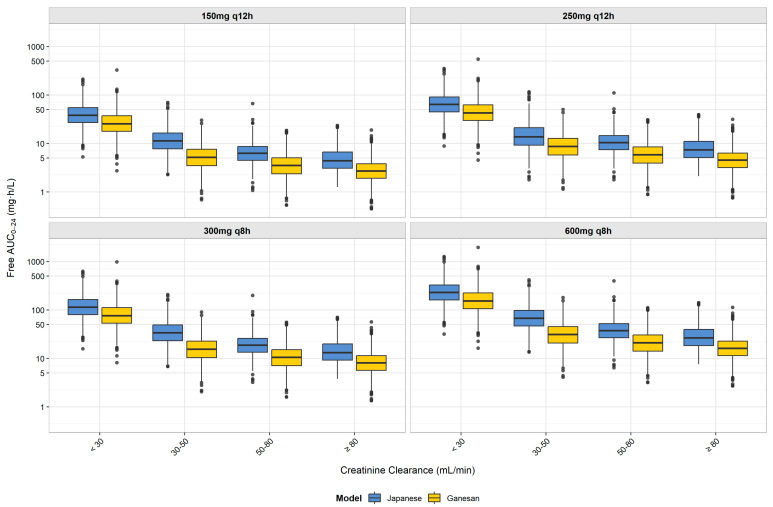
Comparison of free AUC_0–24_ distribution between Japanese and Ganesan population pharmacokinetic models for tebipenem by renal function group and dosing regimen. Box plots represent the distribution of free AUC_0–24_ values from Monte Carlo simulations (*n* = 1000 per group) for Japanese (blue) and Ganesan (yellow) models. Each box shows the interquartile range (IQR), with the median indicated by the horizontal line, whiskers extending to 1.5 × IQR, and the outliers shown as individual points. Results are displayed by dosing regimen (150 mg q12 h, 250 mg q12 h, 300 mg q8 h, and 600 mg q8 h) for each creatinine clearance group (<30, 30–50, 50–80, and ≥80 mL/min). The y-axis is presented on a logarithmic scale to better visualize the differences in exposure across the wide range of AUC_0–24_ values. The figure demonstrates higher AUC_0–24_ values in the Japanese model compared to the Ganesan model across all renal function groups. These differences in pharmacokinetic profiles support the distinct dosing recommendations between Japanese and foreign patients, with implications for the PTA against ESBL-producing Enterobacterales with various MICs.

**Table 1 antibiotics-14-00648-t001:** Recommended tebipenem dosing regimens for extended-spectrum β-lactamase (ESBL)-producing *Escherichia coli* and *Klebsiella pneumoniae* with a minimum inhibitory concentration (MIC) of 0.03 mg/L.

CCR	Optimal Dosage
CCR ≥ 80 mL/min	600 mg q8 h
50 ≤ CCR < 80 mL/min	300 mg q8 h
30 ≤ CCR < 50 mL/min	300 mg q8 h
CCR < 30 mL/min	150 mg q12 h

CCR, creatinine clearance.

**Table 2 antibiotics-14-00648-t002:** Comparison of the probability of target attainment between Japanese and Ganesan population pharmacokinetic models for tebipenem by renal function group (MIC 0.03 mg/L).

	Japanese Model PTA (%)	Ganesan Model PTA (%)
	CCR < 30 mL/min	
150 mg q12 h	97.3	89.1
250 mg q12 h	99.9	98.9
300 mg q8 h	100	99.9
600 mg q8 h	100	100
30 ≤ CCR < 50 mL/min		
150 mg q12 h	40.6	5.4
250 mg q12 h	74.9	24.2
300 mg q8 h	99.4	84.7
600 mg q8 h	100	98.7
50 ≤ CCR < 80 mL/min		
150 mg q12 h	7.4	1.2
250 mg q12 h	32.3	9.6
300 mg q8 h	93.3	62.2
600 mg q8 h	99.7	93.3
CCR ≥ 80 mL/min		
150 mg q12 h	3.2	0.2
250 mg q12 h	17.2	2.2
300 mg q8 h	79.4	45.3
600 mg q8 h	98.4	87.9

CCR, creatinine clearance; PTA, probability of target attainment.

**Table 3 antibiotics-14-00648-t003:** PK model, parameters, and creatinine clearance-based parameter values used for MCS.

Pharmacokinetic Model		Final Model		
One-compartment model with first-order absorption		Pharmacokinetic parameters		
		CL/F (L/h), Vd/F (L), ka (h^−1^), fu		
		ωCL/F (%) = 53		
		ωVd/F (%) = 37		
		ωka (%) = 67		
**Pharmacokinetic Parameters**	**CCR ≥ 80**	**50 ≤ CCR < 80**	**30 ≤ CCR < 50**	**CCR < 30**
CL/F (L/h)	21.738	16.176	8.61	2.724
Vd/F (L)	26.461	34.352	17.906	15.754
ka (h^−1^)	2.5	1.1	2.8	1.7
Fu	0.33	0.33	0.33	0.33

CL/F, apparent clearance; Vd/F, apparent volume of distribution; ka, absorption rate constant; fu, fraction unbound; CCR, creatinine clearance (mL/min).

## Data Availability

All applicable data are contained in the paper.

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
