# Peer review of "Preclinical Study of Pharmacokinetic/Pharmacodynamic Analysis of Tebipenem Using Monte Carlo Simulation for Extended-Spectrum β-Lactamase-Producing Bacterial Urinary Tract Infections in Japanese Patients According to Renal Function"

_antibiotics, 2025, doi:10.3390/antibiotics14070648_

Round 1
Reviewer 1 Report
Comments and Suggestions for Authors
This manuscript explores the optimization of tebipenem dosing for ESBL-producing urinary tract infections in Japanese patients. The authors employ PK/PD modeling via Monte Carlo simulations, which is methodologically sound and provides a valuable approach to address this therapeutic challenge. That said, a few key aspects require deeper clarification to enhance the manuscript’s rigor and transparency. In particular, the researchers should provide a more detailed justification for applying pharmacokinetic variability data from faropenem instead of using tebipenem-specific data. data. Overall, the presentation of results is clear and supported by well-constructed tables and figures. Table 1 stands out as especially informative. However, the manuscript would benefit fvrom including a direct discussion on the practicality of q8h oral dosing in outpatient care.Additionally, while the inclusion of international data enhances the manuscript, it could more clearly articulate its implications for clinical practice by providing specific examples of how different settings can apply this data. In conclusion, the manuscript is well-written and addresses a timely topic. With a few targeted revisions, primarily to improve explanatory depth and contextual nuance, it has the potential to make a meaningful contribution to the literature.
Reviewer 2 Report
Comments and Suggestions for Authors
To highlight the clinical significance of ESBL-producing organisms in UTI, it may be helpful to include epidemiologic data on the burden of ESBL-related infections, preferably from Japan or, if unavailable, from global surveillance data.
If available, please include the approved and maximal dosing information for TBPM in pediatric patients in Japan.
For dialysis-dependent patients, additional information on whether TBPM is excreted via HD or PD would be valuable. If possible, please also suggest a dosing strategy for such populations.
